# Muscle activation and intermuscular coordination adaptations to early strength training during maximal force production

**Paulo D. G. Santos**[1,2]*, **João R. Vaz**[3], **Miguel Gomes**[1], **Jorge Infante**[4],
**Pedro Pezarat-Correia**[3]

**1** Faculty of Human Kinetics, Neuromuscular Research Lab, Lisbon, Portugal, **2** Faculty of Human Kinetics, CIPER, Lisbon, Portugal, **3** Egas Moniz Centre for Interdisciplinary Research (CiiEM), Egas Moniz Health School of Science, Monte da Caparica, Portugal, **4** Faculty of Human Kinetics, SpertLab, Lisbon, Portugal

* pauloduarteguiasantos@gmail.com

## Abstract

Previously untrained individuals tend to increase maximal and rapid strength during the initial resistance training stages. Muscle activation and coordination are neural mechanisms contributing to the increased mechanical output. However, how the force production characteristics are accompanied by changes in activation of agonist muscles as well as coordination of muscles with different functional roles is not fully understood. This study investigated the time course of adaptations following 6 weeks of resistance training, evaluating every two weeks a leg-press isometric maximum voluntary contraction. Peak force (PF), rate of force development (RFD), rate of EMG rise (RER) of the agonist muscles and intermuscular coherence between synergist or antagonist pairs of muscles were evaluated. In result of a dynamic squat program, the maximal and rapid leg-press isometric force increased after the 6-week period ($p = 0.011$ and $p = 0.015$, respectively), although improvements at specific intervals of RFD at specific time points were observed. Regarding knee extensor activation, generally decreased RER was observed only for rectus femoris and not for the monoarticular portions of quadriceps. Additionally, intermuscular coherence analysis revealed increased coupling between rectus femoris and the monoarticular portions of quadriceps after training, and adaptations between agonist muscles acting in different joints as well as between agonist and antagonist muscle at specific time points were observed concerning specific bands. This is the first study to characterize the time course of intermuscular coordination adaptations during the early phase of strength training in previously untrained individuals, bringing new insights into the neural mechanisms of muscle recruitment following resistance training in what concerns to coordinative strategies in the control of muscles with different functional roles.

**Data availability statement:** We confirm that the manuscript reports all summary statistics and inferential outcomes used to support the study findings. The underlying de-identified individual-level data required to reproduce the reported analyses (i.e., values underlying means, standard deviations, and figures) are available from the following Zenodo link: https://doi.org/10.5281/zenodo.18630339.

**Funding:** [Fundação para a Ciência e Tecnologia, Lisboa, Portugal, under grant number SFRH/BD/146411/2019 (https://doi.org/10.54499/SFRH/BD/146411/2019)]. The funders had no role in study design, data collection and analysis, decision to publish, or preparation of the manuscript.

**Competing interests:** The authors have declared that no competing interests exist.

## Introduction

Strength training induces increased capacity of force production that emerges from neuromuscular adaptations. Previously untrained individuals present lower maximal [1,2] or rapid force [3–5] than trained ones, which has been shown to be related to muscular adaptations, such as changes in muscle size [6,7] and architecture [8–10], but also to neural factors. More precisely, strength trained individuals have shown to be capable of producing greater voluntary activation of muscles during maximal force production [1,2,4,11] presenting adaptations spanning from the cortex to the spinal cord, as reduced intracortical inhibition and changed corticospinal pathway [12,13], or motor unit recruitment with increased discharge rate [14,15].

During the first weeks of training, adaptations seem to be mainly related to neural mechanisms, while muscular adaptations present greater relevance after 6–8 weeks of training [16,17]. For example, during this initial training stage, increased cortico-spinal excitability and reduced short-interval cortical inhibition as well as reduced activation of the antagonist muscles were observed after only 2 weeks of isotonic wrist flexion training [18]. Also, changes in the activation of stabilizers during elbow flexions after 3 weeks [19], increased voluntary activation and rate of activation of the plantar flexors [20] or decreased recruitment threshold of motor units and increased discharge rate [15] after 4 weeks were observed. However, how these specific adaptations emerge overtime is still not entirely understood, and it may be crucial to a more effective strength training prescription. For example, different methods of strength training have demonstrated to promote different acute adaptations in muscle activity assessed by electromyography (EMG) in groups with different training background [21] which may lead to distinct long-term adaptations. Additionally, strength adaptations of untrained individuals during initial strength training stages seem to be influenced by the pre-existing strength level [22].

The use of heavy loads is recommended to increase maximal strength [23,24]. Additionally, it has been suggested that training programs with heavy loads promote increases not only in maximal capacity of force production but also in rapid force production measured by rate of force development (RFD) which is the rate at which the force-time curve increases in explosive contractions [25–27]. Also, when analyzing specific time points of this curve, it is possible to distinguish the origin of adaptations. On one hand, the initial 50–100ms have been related to neural adaptations as greater firing frequency [14,15,28,29] or greater rate of agonist muscle activation [4,5,30,31]. On the other hand, muscular adaptations as muscle thickness, cross-sectional area and muscle architecture [31–34] seem to be associated to adaptations in late RFD.

As previously described, although several studies have characterized the input from the Central Nervous System (CNS) to the agonist muscle during strong contractions, few studies have evaluated the intermuscular coordination patterns during force production. Changed agonist-antagonist relationship has been demonstrated following a period of strength training, with the antagonist muscle usually decreasing its activity [1,35,36] although it depends on the elicited task and the mode of contraction [11]. Furthermore, global coordination has been assessed through the extraction of muscle synergies, namely with the comparison of individuals with and without strength training

experience [37,38], although only one study has observed the effects of a strength training program [39]. Nevertheless, other approach has been used to assess specific neural mechanisms of motor control between pairs of muscles, the intermuscular coherence (IMC). IMC quantifies the degree of synchronization between EMG signals from different muscles at the same frequency, reflecting shared neural input or functional connectivity within the neuromuscular system [40–42]. IMC is typically computed as the cross-spectrum between two EMG signals and captures linear, iso-frequency coupling, rather than interactions across different frequency bands. As such, IMC provides information on frequency-specific common drive between muscles. Although frequency bands in EMG coherence analyses are often labelled using terminology derived from electroencephalography (EEG) and particularly corticomuscular coherence research (alpha-, beta- and gamma-bands), these bands do not reflect the same neurophysiological basis as cortical oscillations. Instead, in the context of IMC, these frequency ranges are used as functional descriptors that index frequency-specific common neural inputs to motor neuron pools arising from spinal and supraspinal sources [40,41,43]. Their use is based on empirical observations showing that different EMG frequency ranges are associated with distinct origins of neural drive and motor control strategies, rather than on a direct transposition of cortical rhythms. Nevertheless, strength trained individuals have presented lower IMC in beta band (15–35 Hz) between synergist muscles [44] as well as lower corticomuscular coherence [45]. Additionally, one study evaluating the adaptations in IMC between synergist muscles with 14 weeks of strength training showed that during submaximal contractions at 70% of maximum voluntary contraction (MVC), increased IMC in gamma band (> 40 Hz) may be found, as well as sporadic peaks at MVC [46]. Concerning the relationship between agonist and antagonist muscles, strength trained individuals showed higher suppression of corticomuscular oscillations in beta band than endurance trained ones [47], although there is lack of research around this topic.

Current research still lacks to describe the adaptations with strength training in agonist-antagonist IMC during maximal force production, and even relationships between synergist muscles are not fully understood. Additionally, the timings of these adaptations during the initial stages of strength training are not known, and it may be useful to understand how different mechanisms contribute differently overtime. Thus, the main purpose of this study is to establish the time course of adaptations in muscle activation and coordination in maximal and rapid strength capacity of untrained individuals during the initial stage of strength training. To characterize the neural mechanisms involved in changes concerning maximal and rapid force production, muscle activation and coordination measured by IMC between synergist as well as antagonists during a maximal isometric task will be assessed. We hypothesize that maximal and rapid force production will increase with training and that the associated mechanisms will likely be explained by an increased initial activation of knee extensor muscles as well as changed IMC between muscles with different functional roles, which may reflect an adaptation in global motor control during maximal force production.

## Methods

### Participants

Eleven participants were included in this study (age: 22.6±3.1 years; height: 176.4±7.9 cm; weight: 70.5±14.9 kg). The participants were healthy men without history of musculoskeletal injuries within the last 6 months. Although all participants were active physical education students, they did not have any resistance training background. All participants provided informed written consent and the recruitment took place between September 2022 and July 2023. This study was approved by the Faculty's Ethics Committee (CEIFMH N.º: 24/2021) and all procedures adhered to the Declaration of Helsinki.

### Data collection

**Experimental approach.** Aiming to assess the effect of resistance training in the ability of the neuromuscular system to generate force and to assess the time course of intermuscular coordination adaptations, participants were included in a high-bar back-squat training program.

Each participant was requested to go to the laboratory on six occasions. The first session was used to familiarize the participants with the testing tasks as well as with the back-squat exercise. The remaining five sessions were similar between them and were used to evaluate the mechanical and neuromuscular outputs. The first of these sessions took place one week after the familiarization session and was used as baseline (BSL), after which the participants were instructed to maintain their regular physical activity without adding resistance training exercises. After three weeks of resistance training abstinence, the participants were evaluated again, with this session accounting as the beginning of the training period (W0). After W0, the participants started the training program and were evaluated every two weeks. Thus, the third, fourth and fifth evaluation sessions were conducted after two (W2), four (W4) and six weeks of training (POS), respectively. The evaluation session after the sixth week was the final session that evaluated training adaptations corresponding to the training program end.

**Resistance training.** The training program lasted six weeks with three training sessions per week, which resulted in 18 training sessions. The resistance training program consisted in 5 sets of 10 repetitions with 2 minutes of rest between sets [48] of back-squats. The exercise intensity was set at 75% of 1 repetition maximum (1RM) of the back-squat. The participants were instructed to perform the exercise with the concentric phase following a controlled eccentric phase. In the final repetitions of the last sets the researcher could spot the participants, and in sets at which the researcher help was needed in 2 or more repetitions, the load was decreased by 2,5 kg for the subsequent set. During the training program the relative load was maintained, with absolute load progressions every two weeks relatively to the back-squat 1RM determined during the evaluation sessions.

**1RM Determination.** For the squat 1RM determination, we used a progressive loading squat test. The squat was performed with a high-bar position, and a squat depth corresponding to a knee angle of 80–90º. Participants started to perform 5 repetitions with the unloaded barbell (20 kg), incrementing 10–20 kg each time. Afterwards, participants performed 3 repetitions with each load until the load was challenging, and finally, they performed sets of 1 repetition with smaller load increases (~5 kg) until the RM determination.

**Testing procedures.** The evaluation session started with a 5-minute cycle ergometer (Monark Exercise AB, Sweden) warm-up. Subsequently, participants were evaluated in isometric maximal bilateral leg press (extension of the hip, knee and ankle). A horizontal leg press machine with custom-build force platform (Faculty of Human Kinetics) was used to assess lower limb maximal strength. The leg press machine was adjusted so the participants knee-angle was approximately 110º (107º according to Peltonen and colleagues [49]). Three MVC were performed by each participant. Participants were asked to perform each repetition as fast and forceful as possible and to maintain the maximal force during 3-5s. Repetitions presenting a visible initial countermovement were excluded and one more repetition was performed. Thereafter, the 1RM of the participants was assessed. All strength tests were accompanied by verbal encouragement. The order of testing procedures was kept constant throughout all sessions and participants. Participants were blinded to the specific testing hypotheses, although they would expect that the inclusion in a strength training program would lead to improved strength.

**EMG collection.** All tests were accompanied by data collection of myoelectrical signals. These signals were recorded on 7 muscles of the right side of the body: gluteus maximus (GM), vastus lateralis of quadriceps (VL), rectus femoris of quadriceps (RF), vastus medialis of quadriceps (VM), biceps femoris long head (BF), lateral gastrocnemius (GL) and tibialis anterior (TA). The muscles were selected so we could assess the lower limb agonist muscles during the triple extension of the lower limb, and so that we could assess agonist-antagonist pairs of each lower limb joint. The electrodes were placed according to SENIAM (Surface EMG for Non-Invasive Assessment of Muscles). Before the placement of the electrodes, the skin was shaved and cleaned with alcohol to minimize skin impedance. Surface EMG was acquired using 7 bipolar surface electrodes (EMG Delsys, TrignoTM), fixed with specially designed adhesive interface, aligned with the muscle fibers [50] and then additionally fixed with tape to avoid movement artefacts. Bipolar surface EMG was used instead of other EMG configurations as monopolar electrodes, as it is widely adopted in IMC studies evaluating

strength-related tasks, offering a practical balance between signal quality and spatial reliability, while facilitating consistent electrode placement in longitudinal designs involving repeated neuromuscular assessments. EMG signals were preamplified and band-pass filtered between 10 and 850 Hz, while digitized at 1000 samples/s. After, the EMG signals were corrected for the 48-ms delay inherent to the Trigno EMG system.

## Data processing

**Force data.** Initially, force data was low pass filtered (12 Hz, 4th order Butterworth). The leg press isometric MVC peak force (PF) was defined as the highest value of force achieved in each repetition. Absolute RFD for the leg press MVC was calculated as the slope of the force-time curve (Δforce/ Δtime) in incremental epochs of 50ms (0–50ms; 50–100ms; 100–150ms; 150–200ms) starting from the onset of contraction. The onset of contraction was defined as the time point where the force exceeded the baseline by 10 N. Peak RFD (PRFD) was defined as the maximum force-time slope at time windows of 20ms [51]. At each time interval where RFD was computed, we also calculated the normalized RFD to MVC to assess the percentage of force exerted at each time point. We selected and analyzed the repetition with the highest PRFD if the PF value was > 95% of the highest PF value across the three repetitions.

**EMG processing.** Raw EMG signals of VL, RF and VM were band-pass filtered (20–450 Hz), rectified and smoothed with a low-pass filter (12 Hz, 4th order Butterworth). Normalization was made to the average value of the 100ms across the EMG peak of the leg press isometric MVC. Rate of EMG rise (RER) of VL, RF and VM was determined as the EMG increase calculated from the onset of electrical activity to the point correspondent to PRFD [20]. RER in time intervals of 0–50ms and 50–100ms was also calculated. The onset of electrical activity was defined as the time point where the EMG exceeded the baseline by 3 standard deviations.

**Intermuscular coherence (IMC).** Coherence analysis was performed using Matlab 2018a (Mathworks, Natick, MA, United States). Before data analysis, all data files were visually inspected to detect possible artifacts. In the cases where signals demonstrated some disruption or noise the repetitions were excluded from the analysis. IMC analysis was performed during 2500ms of each repetition, starting from 500ms after the onset of force production. A total of 7.5 seconds were analyzed. Before IMC analysis, raw EMG signals were full wave rectified [46,52,53], and data was divided into non-overlapping windows of 500 samples in length. The signals were subjected to fast Fourier transform, which gave a frequency resolution of 2 Hz. IMC was calculated as the cross-spectrum normalized by the auto-spectrum:

$$C_{xy}(f) = \frac{\left|P_{xy}(f)\right|^2}{P_{xx}(f)P_{yy}(f)}$$

(1)

where $C_{xy}(f)$ is the magnitude squared coherence between two rectified EMG signals, $x(t)$ and $y(t)$, for a given frequency $f$. $P_{xy}(f)$ is the cross-spectrum and $P_{xx}(f)$ and $P_{yy}(f)$ are the auto-spectrum of $x(t)$ and $y(t)$, respectively. Coherence above $Z$ was considered significant [46,54]:

$$Z = 1 - \alpha^{\frac{1}{(L-1)}}$$

(2)

where $L$ is the total number of sections and the significance level α was set to 0.05. The $L$-value was 15.

IMC was computed for the follow pairs of muscles: VL-VM, VL-RF, VM-RF, VL-GM, VL-GL, VL-BF, RF-BF and GL-TA. Total coherence between 10–100 Hz was averaged across participants at each timing of evaluation, as well as coherence sectioned into 15–35, 40–60 and 60–100 Hz bands. Within 15–35 Hz band representing beta band, two additional 15–24, 25–35 Hz bands were separated. Average coherence within each band was used to compare the evaluation timings (BSL, W0, W2, W4 and POS).

## Statistics

Statistical analysis was made using jamovi software version 2.3.28.0 (Sydney, Australia). We verified normality through the Shapiro-Wilk test. A repeated measures ANOVA with four levels (W0, W2, W4 and POS time-points) was used to assess differences during the time course of the training program. When normality was not assumed, a Friedman test was used, and when sphericity was not assumed, a Greenhouse-Geisser correction was made. Tukey adjustments were used in *Post hoc* tests of ANOVA, and Durbin-Conover adjustments were used in *Post hocs* of Friedman test. The previously described analysis was made to all variables (PF, PRFD, absolute and normalized RFD in time intervals, RER of each knee extensor muscle as well as average coherence within each band). As IMC values within each band were not normally distributed we applied a log 10 transformation to allow parametric tests to be performed.

We assessed the relative reliability of the measurements between the BSL and W0 time points by calculating two-way mixed-effects intraclass correlation coefficients [ICC(3,1)] [55], with 95% confidence interval. ICC values were considered: poor, $< 0.5$; moderate, 0.5–0.75; high, 0.75–0.9; excellent, $> 0.9$. Additionally, we calculated standard error of measurement [SEM] as an absolute reliability measurement [56].

All statistical analyses were performed using a significance level of $p < 0.05$. Data is presented as mean $\pm$ SD. Cohen' $d$ was calculated as measure of effect size. Effect size was considered small, medium or large for Cohen' $d$ of 0.2–0.5, 0.5–0.8 and $> 0.8$, respectively. A post-hoc power analysis was conducted (G*Power 3.1.9.4, Düsseldorf, Germany) using the observed effect sizes of the primary outcomes showing differences from W0 to POS, using two-tailed parameters at $\alpha = 0.05$. The achieved power was 0.805 for PF, 0.756 for PRFD, and 0.771 for RER of the RF. For the primary IMC outcomes presenting significant changes with training, the achieved power was 0.820 for TA-GL total IMC, and 0.727 for VL-RF at 15–24 Hz band. These values indicate that the study had adequate sensitivity to detect large effects, while small or moderate effects, especially concerning several IMC variables, may have been underpowered.

## Results

### Reliability

Most mechanical variables regarding force production (PF, PRFD, absolute RFD at time intervals and %MVC at 50ms) revealed high to excellent reliability between BSL and W0 ($0.85 < ICC < 0.98$), with exception of %MVC at 100ms that was moderate (ICC = 0.73) and poor at 150 and 200ms (ICC = 0.37 and 0.29, respectively). The SEM of the variables with high to excellent reliability was consequently low, with the SEM of %MVC at 100, 150 and 200ms, that presented poor to moderate reliability varying from 4.69 to 7.34%.

Regarding muscle activation, there were variables presenting high (RER of RF, RER of RF between 0 and 50ms and between 50 and 100ms), moderate (RER of VL and VM, RER of VL and VM between 0 and 50ms) and poor (RER of VL and VM between 50 and 100ms) relative reliability. The SEM values for RER of VL, VM and RF were similar ranging from 0.07 to 0.08%/ms, and the RER at time intervals for the three muscles ranging from 0.12 to 0.18%/ms.

For the IMC analysis, ICC values revealed that most of the 48 variables (8 pairs of muscles x 6 bands of interest) present moderate to excellent reliability (moderate, 42% of the variables; high, 14%; excellent, 5%), although 39% of the variables present poor reliability.

### Force production

A time effect was shown for PF during the leg press, with the final moment of evaluation (POS) being significantly different from the first (W0; $p = 0.011$ and $d = 0.944$). Although there was a trend for a progressive increment of PF along the weeks, there were not significant differences in post hoc comparisons between W0 and W2 or W4 even with medium effect sizes ($p = 0.064$; $d = 0.516$ and $p = 0.073$; $d = 0.583$, respectively). Concerning PRFD, only differences between W0 and POS were found ($p = 0.015$ and $d = 0.888$). Absolute and relative RFD only presented differences in post hoc comparisons

between W2 and POS for absolute RFD between 50 and 100ms that was greater after training ($p = 0.048$; $d = 0.302$), and in the overall ANOVA for the relative RFD represented by %MVC at 200ms ($p = 0.013$; $d = 0.917$) that tended to reduce with training. Results are detailed in Table 1.

### Knee extensor activation

Muscle activation measured by RER from the onset of electrical activity to the point at PRFD showed to be unchanged for VM and VL. For RF, the ANOVA revealed a significant decreasing effect of training overtime ($p = 0.045$; $d = 0.777$) with post hoc comparisons presenting differences between W0 and W4 ($p = 0.030$; $d = 0.904$). Regarding RER at time intervals for VM and VL there was not any significant training effect. For RF, although there were not significant overall differences, there was post hoc differences between W0 and W2 for RER between 0 and 50ms ($p = 0.044$; $d = 0.904$) and between W0 and W4 for RER between 0 and 50ms ($p = 0.020$; $d = 1.005$) as well as between 50 and 100ms ($p = 0.028$; $d = 0.863$), with all the significant differences representing a decreased activation. Results are detailed in Table 1.

### Intermuscular coherence

Fig 1 shows all the muscle pairs analyzed at each time point. A visual inspection of the coherence data series revealed that IMC at 25–45 Hz, encompassing the higher beta band as well as the initial frequencies of gamma band, presented

**Table 1. Force and muscle activation related variables.**

| Variable | W0 | W2 | W4 | POS |
|---|---|---|---|---|
| **Force Production** | | | | |
| PT (N) * | 2514 ± 578 | 2885 ± 836 | 2877 ± 705 | 2964 ± 664 # |
| PRFD (N/ms) | 9.98 ± 2.13 | 9.94 ± 0.92 | 10.30 ± 1.97 | 11.20 ± 2.70 # |
| RFD0–50ms (N/ms) | 5.36 ± 1.82 | 5.30 ± 1.56 | 5.05 ± 2.13 | 5.78 ± 2.84 |
| RFD50–100ms (N/ms) | 8.97 ± 2.01 | 8.96 ± 1.49 | 9.05 ± 2.05 | 9.60 ± 2.60 § |
| RFD100–150ms (N/ms) | 8.63 ± 2.00 | 8.87 ± 1.02 | 8.96 ± 1.60 | 9.57 ± 2.47 |
| RFD150–200ms (N/ms) | 6.14 ± 2.67 | 6.41 ± 1.67 | 6.10 ± 2.43 | 6.93 ± 3.40 |
| %MVC50ms | 12.28 ± 4.23 | 10.55 ± 4.09 | 9.91 ± 4.61 | 11.27 ± 5.40 |
| %MVC100ms | 31.53 ± 8.93 | 27.41 ± 8.90 | 26.58 ± 9.80 | 28.73 ± 10.80 |
| %MVC150ms | 49.58 ± 10.09 | 43.68 ± 11.06 | 42.75 ± 11.36 | 45.67 ± 11.64 |
| %MVC200ms * | 62.07 ± 9.64 | 55.03 ± 11.60 | 53.43 ± 10.97 | 57.51 ± 10.01 |
| **Muscle Activation** | | | | |
| RER_VL (%/ms) | 0.43 ± 0.08 | 0.39 ± 0.16 | 0.41 ± 0.09 | 0.43 ± 0.12 |
| RER_RF (%/ms) * | 0.36 ± 0.13 | 0.29 ± 0.11 | 0.26 ± 0.08 ### | 0.29 ± 0.09 |
| RER_VM (%/ms) | 0.44 ± 0.08 | 0.41 ± 0.13 | 0.43 ± 0.16 | 0.43 ± 0.19 |
| RER_VL0–50ms (%/ms) | 0.75 ± 0.17 | 0.65 ± 0.37 | 0.70 ± 0.22 | 0.76 ± 0.30 |
| RER_RF0–50ms (%/ms) | 0.73 ± 0.31 | 0.46 ± 0.26 ## | 0.44 ± 0.24 ### | 0.51 ± 0.21 |
| RER_VM0–50ms (%/ms) | 0.79 ± 0.17 | 0.61 ± 0.32 | 0.73 ± 0.37 | 0.74 ± 0.42 |
| RER_VL50–100ms (%/ms) | 0.74 ± 0.24 | 0.61 ± 0.25 | 0.74 ± 0.19 | 0.66 ± 0.19 |
| RER_RF50–100ms (%/ms) | 0.70 ± 0.24 | 0.53 ± 0.23 | 0.48 ± 0.26 ### | 0.60 ± 0.18 |
| RER_VM50–100ms (%/ms) | 0.70 ± 0.17 | 0.67 ± 0.22 | 0.65 ± 0.21 | 0.55 ± 0.28 |

EMG – Electromyography; MVC – Maximum voluntary contraction; PF – Peak force; POS – Ending of training; PRFD – Peak rate of force development; RER – Rate of EMG rise; RF – Rectus Femoris; RFD – Rate of force development; W0 – Training beginning moment; W2 – Week 2 of training; W4 – Week 4 of training; VM – Vastus Medialis; VL – Vastus Lateralis. * $p < 0.05$ in overall repeated measures ANOVA; # $p < 0.05$ in post hoc comparisons between W0 and POS; ## $p < 0.05$ in post hoc comparisons between W0 and W4; ### $p < 0.05$ in post hoc comparisons between W0 and W2; § $p < 0.05$ in post hoc comparisons between W2 and POS.

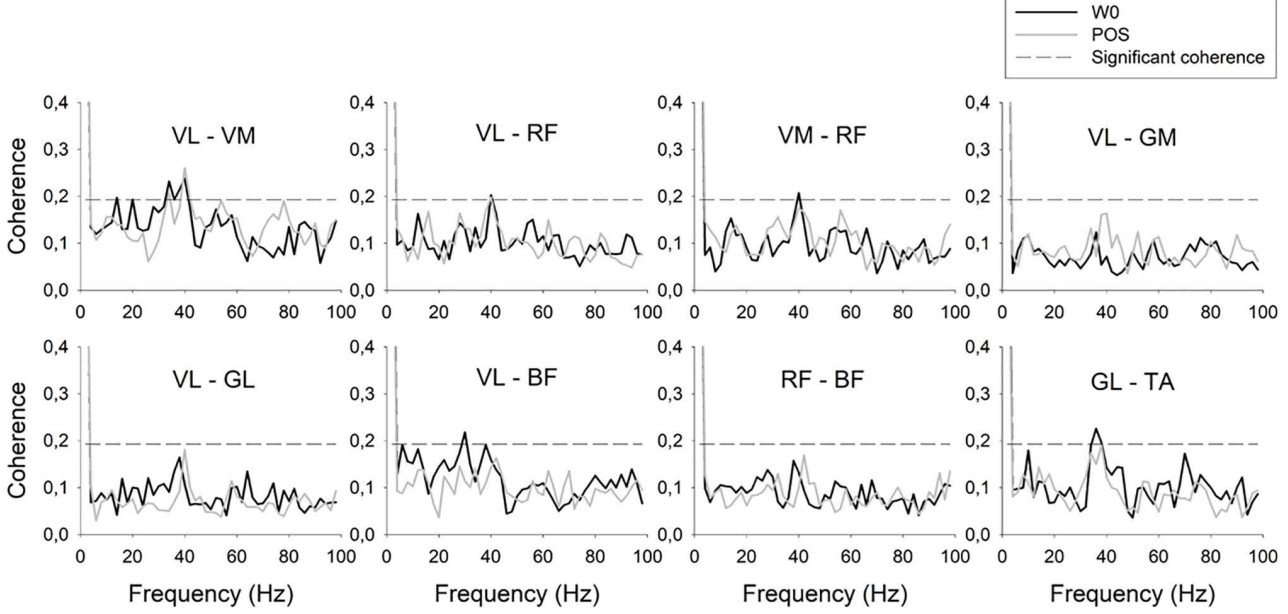

**Fig 1. Intermuscular coherence of the analyzed pairs of muscles at W0 and POS between 0 and 100 Hz.** The horizontal dashed line represents significant coherence, and frequency-specific effects are reported quantitatively in the Results section. BF – biceps femoris; GL – gastrocnemius lateralis; GM – gluteus maximus; POS – Ending of training; RF – rectus femoris; TA – tibialis anterior; VM – vastus medialis; VL – vastus lateralis; W0 – Training beginning moment.

significant peaks between VL and VM. IMC between VM and RF has also presented a significant peak at this frequency only at W0. Significant peaks at occasional time points have been shown between VL and BF at 15–35 Hz bands, and between GL and TA at 25–45 Hz only at the two first evaluations (W0 and W2).

The overall result of the ANOVA for each pair of muscles at each time point did not show any significant effect of training in IMC with exception of GL and TA IMC between 10–100 Hz that tended to be reduced with training ($p = 0.025$; $d = 0.963$). However specific differences between time points have been revealed by post hoc comparisons. Concerning agonist-agonist pairs, increased IMC between VL and RF at 15–24 Hz ($p = 0.017$; $d = 0.857$) was shown between W0 and POS. Regarding synergist pairs acting at different joints, VL and GL presented decreased IMC at 25–35 Hz after the six weeks of training ($p = 0.047$; $d = 0.784$). Agonist-antagonist pairs have also revealed occasional differences, with decreased IMC between VL and BF after the first two weeks of training at 15–24 Hz ($p = 0.017$; $d = 0.567$) band, although these changes had not been maintained during the subsequent weeks. Additionally, decreased IMC at 10–100 Hz frequency range has been shown between RF and BF between W4 and POS ($p = 0.008$; $d = 0.065$) and between GL and TA between W2 and W4 ($p = 0.048$; $d = 0.725$).

## Discussion

The present study assessed adaptations in lower limb muscle activation and coordination following a 6-week strength training period, describing the time course of those adaptations. Training may lead to adaptations in the mechanics of force production as a result of specific adaptations in muscle activation and coordination strategies. Maximal force production during a leg press task tended to increase during the training period, although PF and PRFD have only increased significantly after the six weeks of training. The assessment of RER revealed reduced RF activation, and the IMC analysis showed significant peaks between pairs of agonists as well as between pairs of agonist-antagonist muscles emerging

after training. Although the IMC reliability was moderate to high for most variables, there were also some variables presenting only poor reliability, and caution should be taken when interpreting results. Nevertheless, there were coupling changes across the analyzed bands (for example, reduced TA-GL total IMC or increased VL-RF at 15–24 Hz), with muscles with different functional roles presenting specific adaptations overtime.

First, it is important to consider the reliability calculations that involved the experimental group as its own control. The mechanical data presented promising reliability values. PF, PRFD, absolute RFD and relative RFD at 50ms presented high to excellent reliability. This is in line with previous research considering the plantar flexor muscles that found also high to excellent reliability with exception for the initial 50ms of contraction that seems to present greater intersession variability [57,58]. In our study we found high to excellent reliability in these early RFD measurements. The automated definition of the onset, as we measured may be more reliable than in studies using manual detection [59]. However, we have just found poor to moderate reliability in relative RFD at time points from 100ms onwards, which may be corroborated by a previous study also assessing the leg press and presenting unreliable measurements [60]. Additionally, the calculation of the mentioned variables considers the PF of the day, and as we used the MVC repetition with the highest PRFD unless it did not reach 95% of the maximum of the day, it may present more variation namely at later intervals. Concerning muscle activation measured from the EMG-time curve, results from poor to excellent reliability have been shown previously, depending on the studied muscle as well as the task [57,60,61] which are in line with our results. Regarding IMC measurements, although most variables presented moderate to high reliability, some variables only presented poor reliability. Previous research has shown that monopolar current measurements may be preferable during tasks without impacts (as the isometric leg-press) or between muscles far enough to avoid crosstalk [62]. Nevertheless, bipolar electrodes, as the ones used in the present study, may be used to study IMC, although they can be less sensitive to detect changes in IMC than the monopolar current systems. Bipolar electrodes also revealed good reliability between vastus lateralis and vastus medialis [62]. Additionally, we cannot discard the hypothesis that untrained individuals may present higher interday variability in IMC.

## Maximal and rapid strength

Six weeks of squat training led to increased PF. Previous research showed increased isometric MVC PF following six weeks of dynamic strength training [24,25,63,64]. Although the sample size is small and caution should be taken when interpreting the results, it is interesting that our results only demonstrated a training effect after the sixth week of training. Previous studies reported increased isometric knee extension PF following only 3 weeks of training but only when training is performed with high loads (80% 1-RM) [24]. The same study showed that PF continued to increase until the completion of the 6 weeks for the group who trained with high loads, and that significant increases in PF with low loads were only achieved after 6 weeks. Others also found increases in isometric knee extension torque/force after 10 days [65] or 3 weeks of dynamic knee extension training [66], in isometric wrist flexion after 3 weeks of biceps curl training [67] or 2 weeks of dynamic wrist flexion training [18]. However, 2 weeks did not seem enough to increase isometric strength following dynamic elbow flexion training, with significant increases being observed after 4 weeks only with high loads [68]. Although significant results were only observed at the final evaluation, a progressive trend to an increase may be observed. This progressive trend to an increase may also be observed after 2 [64] and 3 weeks of strength training [25] although significant results were only achieved after the sixth week. The study of Knight and Kamen [64] has shown that individuals without training background present an increase in isometric force due to familiarization even without training. The familiarization session and the previously BSL session performed in our study may have helped to minimize the magnitude of this learning effect, although BSL and W0 measurements have shown good reliability. Additionally, task specificity may also have influenced the temporal pattern of the observed adaptations. The training program involved dynamic squats, while the testing task was an isometric leg-press. It is well-known that strength gains and the neural adaptations are specific to the training modality (as velocity and load) and contraction type [19,69,70]. Dynamic resistance training has

been shown to transfer only partially to isometric testing, with reduced improvements compared to task-matched training conditions [71]. Therefore, the limited changes observed at intermediate time points may reflect incomplete transfer of training-induced adaptations to the isometric task rather than an absence of neuromuscular adaptations. This consideration is especially relevant when considering intermuscular coordination which is highly task dependent.

We have also found greater PRFD after six weeks of training. Previous research showed increased RFD after 6 weeks of dynamic strength training [25,72]. In our study we did not find differences in 0–50ms interval, but a trend for increased 50–100ms absolute RFD was observed. Previously, the 0–50ms interval in RFD analysis have shown to be greater in strength trained individuals when compared to untrained ones, although the 50–100ms interval may be increased for the untrained individuals possibly because of delayed activation of the agonist muscles [3–5]. These intervals are related to greater neural input to the muscles, namely greater firing frequency of motor units and muscle fiber conduction velocity [3,14]. The fact that we only verified significant results at 50–100ms time window may suggest that 6 weeks led to increased neural output to the muscles, although this activation may be delayed when comparing with long-term trained individuals. Concerning later intervals, our results have also shown decreased relative RFD at 200ms. The later intervals are related with cross-sectional area or muscle thickness, as well as PF [28,31]. Although we do not present muscular adaptations to the 6-week training program, it is plausible that the load dynamics adopted in this study led to gains in muscle mass. The decreased relative RFD at 200ms may be surprising, although it results from an unchanged absolute RFD at 150–200ms time interval, and a great magnitude increase in PF during the training period.

Furthermore, dynamic strength training programs have shown to increase RFD at specific time windows. Increased maximal rate of torque development of knee extensor and flexor muscles was observed after 10 weeks of barbell deadlift training, but also at time windows of 50 and 200ms [73]. Similar results were also achieved after 14 weeks of heavy lower limb training [30]. Also, heavy leg press training (90% 1-RM) during 8 weeks led to increased RFD at early and later time windows when performed with or without eccentric overload [27]. It is important to note that specific results may be found in current research. For example, 8 weeks of bench press training have also increased early RFD at 50ms during a bench press, but only when the velocity loss during the training program was 0%, while greater velocity losses resulted in increased later RFD [26]. Additionally, velocity losses above 20% have shown to decrease the early 50ms RFD in isometric squat following 8 weeks of squat training [74]. One study comparing 1 vs 4 vs 8 exhausting squat sets also found decreased RFD at all time intervals [75]. The previous cited studies observed differences in previously trained men, and as it was stated above, different training background will lead to different neuromuscular adaptations. Also, differences in study design may lead to different results.

## Knee extensor activation

Muscle activation was measured by RER, and it was observed that the monoarticular portions of the quadriceps remained similar from W0 to POS, while RF decreased its activation overtime. This may be surprising, since RF is agonist of knee extension, and it was expected a greater activation of the knee extensors. Previous studies have found increased activation during the initial epochs of the contraction of plantar flexors following 4 weeks of isometric plantar flexion training [20], knee extensors of strength trained athletes during the initial 25ms of contraction [5], elbow flexors during the initial 30ms only after 2 weeks of dynamic elbow flexion training [76], VL during a jump squat after 10 weeks of power training only for a stronger group [22], or knee extensors during isometric knee extension after 14 weeks of heavy lower limb training [30]. The fact that the participants of the present study did not have any strength training experience may be related to the absence of adaptations concerning VL and VM activation, as suggested by James, Gregory Haff [22]. Also, the remaining enumerated studies evaluated single-joint while in our study we have evaluated a multi-joint exercise. This fact may have resulted in different neuromechanical patterns of force production since there is evidence of different neural input to agonist muscles between single- and multi-joint exercises [77] as well as between closed (as the leg press) and open kinetic chain (as knee extension) [78,79]. Indeed, the decreased activation of RF observed in the present study, although

unexpected, aligns with recent evidence concerning the role of biarticular muscles, may be related with better efficiency of RF activation as a biarticular muscle, agonist of knee extension and antagonist of hip extension. First, studies using high-density surface EMG have shown that RF is regionally and task-dependently activated [80,81], also depending on hip flexion angle [82]. Second, the activity of RF during closed chain exercises has been demonstrated to be lower when compared to open chain exercises [77,83]. Third, during multi-joint exercises combining knee extension and hip extension, as the leg-press, RF activation is decreased when compared to single-joint knee extension, which may contribute to increased hip extension torque [77]. Consequently, it has been shown that single-joint knee extension training is preferential to enhance RF activation rather than multi-joint exercises [84]. Therefore, a reduction in RF RER after squat training may reflect a more efficient distribution of the neural drive during the task, reducing RF early activation when knee extension is combined with hip extension, and enhancing coordination with the monoarticular portions of the quadriceps. Nevertheless, recent work from our research group presented a trend for reduced RER of RF in highly trained individuals when compared to untrained ones, with different fatigue-related responses between groups (with the trained individuals increasing and the untrained individuals decreasing RF activation after fatigue) [85]. These findings further support that RF activation patterns are modified with training experience, and therefore the observed reduction in RF RER may reflect a task-specific adaptation to optimize the intermuscular coordination patterns. Future studies using high-density surface EMG should explore the regional adaptations in RF activation to confirm if the observed activation results from a redistribution of the neural drive or an overall neural input to the muscle.

### Intermuscular coherence

One technique to study intermuscular coordination has been through muscle coupling between pairs of muscles with distinct functional roles, and it has been suggested that neural control of motor units of different muscles may not be independent, merging the concept of common input to motoneurons and synergistic coordination of movement [86,87]. Within this conceptual framework, IMC analysis has been used to distinguish signal input frequencies common between a pair of muscles, and concerning the correlation between the respective power spectral densities it allows to infer about the neural origin of movement control [43]. It is important to note that the frequency bands used in EMG-based IMC analysis are not physiologically equivalent to EEG-derived alpha-, beta-, or gamma-bands. Surface EMG reflects the summation of motor unit action potentials, whereas EEG reflects synchronized synaptic activity at the cortical level. Consequently, IMC frequency bands should be interpreted as reflecting frequency-specific common neural input to motor neuron pools from spinal and supraspinal origins, rather than direct cortical oscillatory activity [40,41,43]. The adoption of EEG-like terminology in IMC studies is largely used, reflecting established conventions in the literature, allowing comparison across studies. Nevertheless, interpretations should remain cautious, and future work combining EMG coherence with cortical measures (e.g., EEG–EMG coherence) may further clarify the neural origins of frequency-specific intermuscular coupling.

Concerning agonist coupling, in this study, we identified at W0 significant IMC peaks between 25 and 45 Hz between the single-joint portions of quadriceps and between VM and RF. This frequency range encompasses frequencies of beta (15–35 Hz) and gamma (> 40 Hz) bands usually associated with corticospinal pathway [40,88–91]. Beta band has also been linked to oscillations within the sensorimotor cortex and the integration of information from other cortical and subcortical structures as well as Ia afferents or Renshaw cells [92,93], while gamma band has been associated to attentional focus and efferent drive during strong isometric voluntary contractions or dynamic effort [94–97]. The peaks observed in IMC between VL and VM have remained, although IMC at lower beta band (15–24 Hz) decreased after 6 weeks of training. On the other hand, IMC between VL and RF increased within this frequency range. The fact that rate of activation has decreased but coupling between VL and RF increased at this band may have been an adapting process to better allow the kinetic output transmission from proximal to distal joints. Namely, since RF is antagonist at hip joint, its rate of activation may be decreased to facilitate an increase in torque development at that joint. Concurrently, augmenting common neural input may be more effective to allow a better synchronization with the monoarticular powerful muscles of

quadriceps. It was also observed greater IMC at beta band between VM and RF in strength trained individuals [85] when compared to untrained ones, which means that the greater coupling between RF and the monoarticular knee extensors may be a neural adaptation to optimize maximal force production. On the contrary, one study has found lower beta band IMC between VM and RF in strength trained individuals [44]. However, in this study the method of calculation of IMC was based on wavelet analysis, and the evaluated task was an isometric knee extension which requires different patterns of muscle activation as discussed previously. Furthermore, lower IMC at gamma band was observed between RF and VM or VL than between VM and VL during squats which may suggest a specific mechanism of control of biarticular and monoarticular muscles during dynamic tasks [98]. Nevertheless, no significant changes were observed between VM and VL, which is in line with previous research that did not find differences in IMC between those muscles at any band following 14 weeks of strength training, although greater IMC at gamma band has been found during submaximal contractions at 70% MVC [46]. Furthermore, it is interesting that corticomuscular coherence, measured between electroencephalographic and EMG of lower limb muscles as the BF, TA and soleus, was lower in ballet dancers and weightlifters than in untrained individuals, which reveals a non-synchronized discharge of corticospinal cells in chronically trained athletes, which suggests that long-term training leads to fine postural adjustments dependent on specific temporal sequences [45].

Regarding synergist muscles acting at different joints, we have also found differences concerning the neural input between VL and GL as well as VL and GM, although VL and GM IMC was an occasional event that resulted from increased IMC of three participants. In its turn, VL and GL showed a trend to a decreased IMC at higher beta band (25–35 Hz). As we mentioned before, this band may be associated to corticospinal pathway, and the decreased IMC observed after training may arise from greater efficiency in muscle recruitment to produce joint moments. The fact that we observed altered patterns concerning the integration of RF during the task may also be applied to GL, considering its role as a biarticular muscle, antagonist of knee extension and agonist of plantarflexion. The reduced common corticospinal input to VL and GL after 6 weeks of training may be related to an adaptive role of GL as an ankle stabilizer during lower limb extension tasks [99,100], providing a more efficient force transmission. Additionally, the changes in the shared neural input to both muscles may be the result of a flexible adaptation in the recruitment of muscle synergies [101,102], supporting the beta oscillations from cortex to muscles [103].

Regarding the agonist-antagonist analyzed pairs, GL and TA showed significant peaks between 25 and 45 Hz (encompassing frequencies from beta and gamma bands). General IMC (10–100 Hz) was reduced after the 6 weeks of training. These results may suggest that both muscles share corticospinal projections that tend to be decreased by training, thus reducing co-activation. However, different neural strategies may be adopted concerning the task and the training mode. For example, badminton players presented higher GL and TA synchronization than untrained individuals, although reducing EMG amplitude of the GL during an isokinetic dorsiflexion, which may be interpreted as a more effective control strategy while maintaining ankle stability [104]. Additionally, it has been shown that distal muscles present stronger corticospinal connections than proximal ones [45,105,106] with weaker Renshaw cell inhibitory feedback [107,108] and greater number of muscle spindles in some muscles as TA or soleus compared to RF or BF [45,109], which may suggest that modulation of those mechanisms contributes to adaptations in IMC after training. However, these interpretations are inferential, since the present data refers exclusively to EMG signals and do not provide direct evidence of corticospinal modulation.

We have also observed that pre-training IMC peaks at 15–35 Hz and < 15 Hz band between VL and BF decreased after two weeks of training, although this behavior has not been maintained. This may have been a compensatory neural strategy temporally associated with the early phase of training during which muscle soreness or damage is often reported in untrained individuals. Previous research focusing on adaptations following eccentric training reported that biceps brachii muscle soreness only dissipated after two weeks of training [110]. Although these findings should not be directly transposed to our study, given our concentric training program, and that in the absence of direct perceptual or biochemical measures this interpretation remain speculative, it would be expected that muscle soreness was present in the first weeks

of training and that the respective prolonged fatigue led to altered agonist-antagonist coordination. Namely, decreased IMC at lower beta (15–24 Hz) and upper alpha between 10 and 14 Hz (since the EMG hardware do not allow to fully characterize alpha-band), may be a consequence of lower corticospinal connectivity, but also from afferent feedback associated to alpha band [111–113]. Although our data may not represent the full alpha band, since the used EMG electrodes present a lower band pass filter threshold of 10 Hz, alpha band is linked to physiological tremor, and although its origin is multifactorial, the modulation of afferent feedback appears to be one mechanism contributing to IMC in this band [111]. It has been observed that afferent feedback modulation during an acute fatiguing state affects the CNS control of muscle recruitment [114,115]. Thus, despite further research is needed to clearly support this, the initial weeks of training may have led to prolonged fatigue that elicited specific agonist-antagonist connectivity modulated by the integration of corticospinal and sensory information. On the other hand, the decreased general IMC between RF and BF after W4, suggests that initial neural adaptations after 2 weeks of training may be transitional adjustments until the system becomes more efficient with task-specific optimization of central commands achieved with long-term training [45]. Furthermore, long-term strength trained individuals have shown different mechanisms of motor control of agonist and antagonist muscles, with lower activation and greater corticomuscular beta band suppression of the antagonist muscles [47,116].

Taken together, the direction and magnitude of IMC changes with training were dependent on both muscle pair and frequency-band, suggesting that early resistance training adaptations do not follow an uniform pattern. While increased IMC may reflect enhanced common neural input or improved functional coupling between muscles, decreased IMC may indicate a refinement of motor strategies and reduced shared drive as task execution becomes more efficient. Importantly, these adaptations appear to be frequency- and muscle-specific, highlighting that intermuscular coordination emerges from multiple parallel neural mechanisms rather than a single unifying process.

The results of this study should be interpreted with caution. First, although valid, the general limitations of the applied data processing methods should be considered when confronting the results with previous research. For example, onset definition of force and EMG may have impact in the results concerning RFD or RER, respectively. Second, muscle selection is determinant for IMC analysis results, as well as the evaluated task. Third, further research should consider using EMG electrodes that do not provide an automatic band-pass filter to a better understanding of full power spectrum from 0 Hz. Fourth, the rectification of EMG signals prior to IMC calculation is still a matter of debate. Rectification is highly recommended for low-force contractions [117], although it was not the case of our study. We opted by rectifying EMG signals to a better comparison with relevant studies although both options may be found in the literature. Fifth, the methods of IMC calculation can lead to different conclusions, although most studies calculate magnitude squared coherence as it was done in this study. Sixth, the present study focused on intermuscular coherence, which quantifies linear synchronization between EMG signals at identical frequencies. While IMC provides valuable insights into shared neural drive and functional coupling, it does not capture non-linear or cross-frequency interactions between muscles. Recent methodological developments, such as Amplitude-Amplitude Cross-Frequency coupling approaches, have been proposed to assess transient and non-stationary coordination across distinct EMG frequency bands, showing fatigue-, sex- and age-related differences in patterns of intermuscular coordination [118–120]. These approaches may offer complementary information regarding intermuscular coordination, identifying distinct rhythms embedded in surface EMG patterns. Future studies could combine IMC with cross-frequency coupling analyses to provide a more comprehensive characterization of neuromuscular connectivity during strength training adaptations. Seventh, it is important to consider that some IMC variables presented only poor reliability between BSL and W0, which may represent interday variability of untrained individuals, and that the significant results should be interpreted with caution because of the small sample size and generally low effect sizes in IMC adaptations. Furthermore, the conducted post-hoc power analysis concerning the main mechanical and electromyographic variables indicated adequate sensitivity for detecting large effects, although small and moderate effects may not have been detectable. Finally, another limitation of the study is the absence of a non-training control condition. Although two baseline assessments were performed (BSL and W0) without training between them, and reliability analysis

confirmed that the primary neuromuscular variables were reliable across sessions, we cannot fully exclude the possibility that part of the observed adaptations over the six-week training period reflect learning. Nevertheless, it is also relevant to note that, within the context of the study, learning-related improvements should not be viewed solely as confounding influences. Rather, they represent a fundamental aspect of early neural adaptations to training, in which intermuscular coordination resulting in more efficient task execution is intrinsic to the physiological processes underpinning initial strength gains.

## Conclusion

Six weeks of training led to increased maximal and rapid force production of previously untrained individuals. However, mechanical variables during intermediate evaluations were unchanged, which was probably caused by the different tasks used in evaluation and in training. Increased PF, maximal RFD and specific changes in time intervals were achieved with concomitant changes in neural mechanisms concerning agonist muscle activation as well as lower limb muscles coordination. Activation of RF was reduced after training, which may be caused by its role as a biarticular muscle. This adaptation may have resulted in specific coordination patterns involving the RF as it was observed by changes in IMC between RF and the monoarticular portions of the quadriceps. Additionally, adaptations between synergist muscles acting in different lower limb joints as well as agonist-antagonist control were observed, suggesting that training led to neuromechanical strategies to enhance the capacity of maximal and rapid force production, although IMC findings should be interpreted with caution, particularly for IMC pairs and respective bands presenting small effect sizes. This study brings new insights into the training adaptations of previously untrained individuals and may be useful to understand the specific neurophysiological responses during an initial training stage, namely concerning the interactions between agonist, synergist and antagonist muscles. Further research should focus on changes in coordination with different training methods in populations with strength training background, so it may be possible to understand if these interactions between muscles remain adapting even during later training stages. Additionally, better understanding on how coordination during dynamic complex tasks evolves in temporal and/or spatial components resulting from synergy extraction and EMG profiling analysis is needed.

## Acknowledgments

The authors would like to thank all participants for their time, commitment, and effort throughout the training and testing procedures.

## Author contributions

**Conceptualization:** Paulo Duarte Guia Santos, João R. Vaz.

**Data curation:** Paulo Duarte Guia Santos.

**Formal analysis:** Paulo Duarte Guia Santos.

**Funding acquisition:** Paulo Duarte Guia Santos.

**Investigation:** Paulo Duarte Guia Santos, Miguel Gomes.

**Methodology:** Paulo Duarte Guia Santos, João R. Vaz, Miguel Gomes, Jorge Infante, Pedro Pezarat-Correia.

**Resources:** Jorge Infante, Pedro Pezarat-Correia.

**Software:** Pedro Pezarat-Correia.

**Supervision:** João R. Vaz, Pedro Pezarat-Correia.

**Validation:** Paulo Duarte Guia Santos, João R. Vaz, Pedro Pezarat-Correia.

**Visualization:** Paulo Duarte Guia Santos.

**Writing – original draft:** Paulo Duarte Guia Santos.

**Writing – review & editing:** Paulo Duarte Guia Santos, João R. Vaz, Miguel Gomes, Jorge Infante, Pedro Pezarat-Correia.

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
