## [Decision Letter · Decision Letter 0]

28 Oct 2025

Dear Dr. Santos,

plosone@plos.org. . . . A rebuttal letter that responds to each point raised by the academic editor and reviewer(s). You should upload this letter as a separate file labeled 'Response to Reviewers'.A marked-up copy of your manuscript that highlights changes made to the original version. You should upload this as a separate file labeled 'Revised Manuscript with Track Changes'.An unmarked version of your revised paper without tracked changes. You should upload this as a separate file labeled 'Manuscript'.

We look forward to receiving your revised manuscript.

Kind regards,

Charlie M. Waugh

Academic Editor

PLOS ONE

Journal Requirements:

[Fundação para a Ciência e Tecnologia, Lisboa, Portugal, under grant number SFRH/BD/146411/2019 (https://doi.org/10.54499/SFRH/BD/146411/2019)].

3. Thank you for stating the following in your manuscript:

[This work was conducted at the Neuromuscular Research Lab, Faculty of Human Kinetics of the University of Lisbon, and supported by the Fundação para a Ciência e Tecnologia under grant number SFRH/BD/146411/2019 (https://doi.org/10.54499/SFRH/BD/146411/2019).]

[Fundação para a Ciência e Tecnologia, Lisboa, Portugal, under grant number SFRH/BD/146411/2019 (https://doi.org/10.54499/SFRH/BD/146411/2019)]

5. Please amend the manuscript submission data (via Edit Submission) to include author Paulo D. G. Santos.

6. Please amend your authorship list in your manuscript file to include author Paulo Duarte Guia Santos.

7. Please include your tables as part of your main manuscript and remove the individual files. Please note that supplementary tables be uploaded as separate "supporting information" files".

Reviewers' comments:

Reviewer's Responses to Questions

**Comments to the Author**

1. Is the manuscript technically sound, and do the data support the conclusions?

Reviewer #1: Yes

Reviewer #2: Yes

2. Has the statistical analysis been performed appropriately and rigorously?

Reviewer #1: Yes

Reviewer #2: Yes

3. Have the authors made all data underlying the findings in their manuscript fully available?

Reviewer #1: Yes

Reviewer #2: No

4. Is the manuscript presented in an intelligible fashion and written in standard English?

Reviewer #1: Yes

Reviewer #2: Yes

Reviewer #1: The manuscript addresses an important topic: the neural adaptations and intermuscular coordination mechanisms underlying strength training in untrained individuals. The experimental design (6 weeks of squat training with repeated assessments of leg-press MVC, EMG, and intermuscular coherence) is novel and potentially impactful. However, some methodological and reporting issues should be clarified before publication.

Major Comments

1.Sample Size & Power

Only eleven participants were included. This small sample size limits statistical power and generalizability. Please provide a priori or post-hoc power analysis to justify whether the study is sufficiently powered to detect meaningful changes.

2.Control Condition

The absence of a control group (non-training) makes it difficult to separate training-induced adaptations from learning/familiarization effects. Authors should acknowledge this limitation more explicitly in the Discussion.

3.EMG & IMC Reliability

Although reliability analyses were provided, some IMC variables showed poor reliability (39%). Results should be interpreted with caution, and this should be emphasized more strongly in the conclusion.

Please clarify why bipolar EMG was chosen instead of high-density or monopolar setups, which may yield more reliable coherence measures.

4.Task Specificity

The training task (dynamic squat) differs from the testing task (isometric leg press). This mismatch could explain why intermediate time points showed limited changes. Authors should expand on this task-specificity issue and its implications.

5.Interpretation of RF Activation

The finding of decreased rectus femoris (RF) activation is intriguing but counterintuitive. The explanation provided (biarticular function, efficiency) is plausible but speculative. It would strengthen the manuscript to compare with existing studies involving biarticular muscles in multi-joint tasks.

6.Figures & Tables

Figures (especially IMC data) are difficult to interpret. Please ensure axis labels, frequency bands, and effect sizes are clearly presented.

Table 1 should include confidence intervals in addition to means and SDs.

7.References

Some recent relevant HD-EMG and intermuscular coherence studies (2021–2024) appear missing. Please update the reference list to ensure full coverage of the field.

Minor Comments

•Abstract: Consider clarifying the main novelty more directly (e.g., “first study to map time-course of IMC adaptations during early strength training”).

•Methods: Provide more details on randomization of test order and whether participants were blinded to study hypotheses.

•Data Availability: The statement is appropriate, but please include repository details if possible (e.g., OSF, Zenodo).

•Language: The manuscript is generally well written but could benefit from minor editing for conciseness.

Recommendation

Major Revision : The study is relevant and potentially impactful but requires clarification and strengthening of methodological justification, discussion of limitations, and improvements in figure presentation.

Reviewer #2: This manuscript investigates adaptations in muscle activation and intermuscular coordination (IMC) following six weeks of strength training. The manuscript and results are interesting, and the topic is relevant and within the scope of PLOS ONE. However, the manuscript presents some conceptual, methodological, and interpretative issues that should be addressed before it can be considered for publication.

INTRODUCTION

Line 66:“This method of analysis of EMG signals refers to the cross-correlation between the frequency power spectrum of two muscles…”

The authors’ definition of intermuscular coherence (IMC) could be confusing. IMC evaluates iso-frequency coupling—that is, synchronization between identical frequency components of two EMG signals. It does not quantify cross-frequency interactions. The current description confuses coherence with cross-frequency coupling (CFC).

In this line, recent developments have introduced amplitude–amplitude cross-frequency coupling (ACFC) as a novel method to quantify non-linear, cross-frequency coordination between muscles (e.g., across distinct EMG frequency bands). These methods extend beyond linear IMC analyses and capture transient, non-stationary couplings across frequencies (doi.org/10.1038/s41598-025-08294-7; doi.org/10.1038/s42003-023-05204-3; 10.1007/s11357-024-01331-9. The authors are NOT required to cite all these works, they are for reference.

Suggestion: Clarify that IMC quantifies iso-frequency coupling (linear synchronization) and not cross-frequency coupling. Also this should be discussed as a potential limitation of IMC in the discussion

METHODS

Lines 176 & 191:

The authors mention both band-pass 10–850 Hz and 10–450 Hz filtering. This inconsistency should be corrected. Furthermore, spectral activity above ~250 Hz in surface EMG is typically negligible and dominated by noise. A physiological rationale for using such a wide filter (up to 850 Hz) is needed.

Additionally, filtering above 20 Hz may remove physiologically relevant low-frequency components. The rationale for excluding sub-20 Hz frequencies should be discussed.

Spectral Rationale:

The manuscript assumes that frequency bands used in EEG (alpha, beta, gamma) apply directly to EMG, yet the neurophysiological basis differs. The authors should justify the transposition of cortical frequency bands to peripheral EMG. I understand that this is commonly done in the literature, but it should be further justified here for context.

RESULTS

The results are generally clear, but the IMC findings are complex and difficult to interpret. The authors should consider adding a summary table or schematic figure indicating which frequency bands and muscle pairs showed increased, decreased, or unchanged coherence. This will help the reader follow the numerous post-hoc findings and reduce the impression of ad hoc interpretation.

DISCUSSION

First paragraph and IMC section:

The authors should specify whether “changes” and “peaks” refer to increases or decreases in IMC. Statements like “changes were observed” are too vague and confusing.

Line 484 and surrounding discussion:

The interpretation that IMC changes reflect “more corticospinal connections” is speculative. The data come from surface EMG, not combined EEG–EMG recordings. Please temper this interpretation or explicitly state that it is an inference rather than direct evidence of corticospinal modulation.

Others:

•If the data were band-pass filtered from 20 Hz, how were coherence peaks at 8–14 Hz analyzed?

•The link between IMC changes and muscle soreness or damage is speculative. No perceptual or biochemical markers of soreness were reported.

•Overall, IMC interpretations seem inconsistent across pairs and frequency bands. The discussion often attributes opposing trends (increases or decreases) to unrelated mechanisms without a unifying framework. The addition of a schematic summary or conceptual model (figure/table) would help clarify these trends.

Needed citations:

The discussion repeatedly refers to “common synaptic input” without referencing seminal or recent work on this topic. Previous studies describing common synaptic input and coordination across muscles (doi.org/10.1113/JP283698, doi.org/10.1152/japplphysiol.00587.2022)

.

Reviewer #1: No

Reviewer #2: No

---

## [Author Response · Author response to Decision Letter 1]

13 Feb 2026

GENERAL ANSWER TO BOTH REVIWERS

Reviewer #1: The manuscript addresses an important topic: the neural adaptations and intermuscular coordination mechanisms underlying strength training in untrained individuals. The experimental design (6 weeks of squat training with repeated assessments of leg-press MVC, EMG, and intermuscular coherence) is novel and potentially impactful. However, some methodological and reporting issues should be clarified before publication.

Reviewer #2: This manuscript investigates adaptations in muscle activation and intermuscular coordination (IMC) following six weeks of strength training. The manuscript and results are interesting, and the topic is relevant and within the scope of PLOS ONE. However, the manuscript presents some conceptual, methodological, and interpretative issues that should be addressed before it can be considered for publication.

Response: We sincerely thank the reviewers for their careful reading of the manuscript and for their constructive comments that will allow to enhance the quality of the work. We appreciate the positive assessment regarding the relevance and novelty of our study, concerning the research of neural and specifically intermuscular coordination adaptations during the initial stages of resistance training in previously untrained individuals.

We also acknowledge the reviewers for the report of methodological, conceptual or interpretation aspects that should be clarified before publication. We have revised the manuscript considering the suggestions of the reviewers. In this document we will address each comment point-by-point, describing the modifications in the manuscript. We consider that the insightful feedback of the reviewers clearly improved the quality of the manuscript.

ANSWERS REVIEWER #1

Major Comments

1.Sample Size & Power

?Only eleven participants were included. This small sample size limits statistical power and generalizability. Please provide a priori or post-hoc power analysis to justify whether the study is sufficiently powered to detect meaningful changes.

Response: We thank the reviewer for this important point. We agree that the sample size is modest and may limit generalizability. We performed a post-hoc power analysis for our primary mechanical and electromyographic outcomes. Using the observed paired effect size (Cohen’s d), N = 11, two-tailed, and p = 0.05, the achieved power for the paired W0 vs POS comparisons were calculated. The results are presented in the table below:

Variable Statistical power

Peak Force 0.805

Peak Rate of Force Development 0.756

Rate of EMG Rise 0.771

IMC Total TA-GL 0.820

IMC 15-24Hz VL-RF 0.727

The presented results indicate adequate sensitivity (power between 75.6% and 82.0%) to detect the large training effect in the mechanical and electromyographic variables. This indicates that the study was sufficiently powered to detect the large effects in Peak Force, Peak Rate of Force Development, Rate of EMG Rise and Intermuscular Coherence variables. However, we acknowledge that the study remains underpowered to detect small or moderate effects, especially for intermuscular coherence variables with higher variability. This is now clearly stated in the manuscript.

Added to the Methods section: A post-hoc power analysis was conducted (G*Power 3.1.9.4, Düsseldorf, Germany) using the observed effect sizes of the primary outcomes showing differences from W0 to POS, using two-tailed parameters at α = 0.05. The achieved power was 0.805 for PF, 0.756 for PRFD, and 0.771 for RER of the RF. For the primary IMC outcomes presenting significant changes with training, the achieved power was 0.820 for TA-GL total IMC, and 0.727 for VL-RF at 15-24Hz band. These values indicate that the study had adequate sensitivity to detect large effects, while small or moderate effects, especially concerning several IMC variables, may have been underpowered.

Added to the Discussion section: Furthermore, the conducted post-hoc power analysis concerning the main mechanical and electromyographic variables indicated adequate sensitivity for detecting large effects, although small and moderate effects may not have been detectable

2.Control Condition

?The absence of a control group (non-training) makes it difficult to separate training-induced adaptations from learning/familiarization effects. Authors should acknowledge this limitation more explicitly in the Discussion.

Response: Thank you for this comment. We agree with the reviewer that the absence of a non-training control group limits the ability to fully separate training-induced adaptations from learning or familiarization effects.

However, as part of our study design, we conducted a reliability analysis using two baseline sessions: one performed three weeks before the beginning of the training program (BSL) and another at the beginning of the training program (W0), with absence of resistance training and the maintenance of regular physical activity between these two moments. This allowed us to assess the reliability of the neuromuscular variables before the training period. Generally, the variables presented high reliability, indicating small familiarization effects. However, some electromyographic variables presented lower reliability, which may indicate familiarization effects and/or variable reliability. Therefore, in the absence of a fully independent control group, some degree of learning across the six-week training period cannot be entirely ruled out, particularly for some EMG variables. We have now revised the Discussion section where we discuss the limitations, to explicitly address this limitation and to clarify that future research should incorporate a non-training control condition to better discriminate learning-related adaptations from training-induced adaptations.

Nevertheless, it should be emphasized that because one of the primary aims of the study was to characterize changes in intermuscular coordination, we consider that task learning itself reflects a meaningful component of early neural adaptations to resistance training. Improved task execution and better muscle coordination are therefore expected and crucial to understanding the initial neuromuscular adjustments elicited by training.

Added to the Discussion section: Another limitation of the study is the absence of a non-training control condition. Although two baseline assessments were performed (BSL and W0) without training between them, and reliability analysis confirmed that the primary neuromuscular variables were reliable across sessions, we cannot fully exclude the possibility that part of the observed adaptations over the six-week training period reflect learning. Nevertheless, it is also relevant to note that, within the context of the study, learning-related improvements should not be viewed solely as confounding influences. Rather, they represent a fundamental aspect of early neural adaptations to training, in which intermuscular coordination resulting in more efficient task execution is intrinsic to the physiological processes underpinning initial strength gains.

3.EMG & IMC Reliability

?Although reliability analyses were provided, some IMC variables showed poor reliability (39%). Results should be interpreted with caution, and this should be emphasized more strongly in the conclusion.

?Please clarify why bipolar EMG was chosen instead of high-density or monopolar setups, which may yield more reliable coherence measures.

Response: Thank you for these important comments. We agree that some IMC variables exhibited low reliability (~39%), and that these results should be interpreted with caution. We have now emphasized this point more clearly in the discussion section, and explicitly reinforced it in Conclusions section, stating that IMC outcomes with poor reliability should be viewed as exploratory and interpreted conservatively, although the neural variables showing significant changes over time all presented moderate to excellent reliability.

Add to Conclusions section: … although IMC findings should be interpreted with caution, particularly for IMC pairs and respective bands presenting small effect sizes

Regarding EMG configuration, bipolar surface EMG was selected because it represents the most commonly used approach for IMC analysis in studies evaluating strength-related tasks. While monopolar EMG may improve spatial resolution and, in some cases, IMC reliability, they also introduce practical and methodological challenges (as reduced feasibility in repeated-measures studies or increased cross-talk between muscles near each other). As it was stated in the Discussion section, we recognize the advantages of using monopolar electrodes, but the use of bipolar electrodes to study IMC is also valid. Nevertheless, we have now clarified why bipolar electrodes were chosen instead of monopolar electrodes.

Added to Methods section: Bipolar surface EMG was used instead of other EMG configurations as monopolar electrodes, as it is widely adopted in IMC studies evaluating strength-related tasks, offering a practical balance between signal quality and spatial reliability, while facilitating consistent electrode placement in longitudinal designs involving repeated neuromuscular assessments

4.Task Specificity

?The training task (dynamic squat) differs from the testing task (isometric leg press). This mismatch could explain why intermediate time points showed limited changes. Authors should expand on this task-specificity issue and its implications.

Response: We agree that the mismatch between the training task (dynamic squat) and the testing task (isometric leg press) may influence the time course and magnitude of the adaptations. Strength gains and neural adaptations are well known to be task-specific, particularly during the early phases of training. Several studies have shown that dynamic strength training led to results with larger effects in dynamic strength tasks than in isometric strength tasks, as described in a recent review (10.1007/s40279-025-02225-2). Additionally, the load- and velocity-specific adaptations of strength training have also been reported, with larger and earlier gains typically observed when testing tasks are close to the training modality and contraction type (10.1007/BF00422902 and 10.2165/11538500-000000000-00000).

In this context, the limited changes observed at intermediate time points may reflect incomplete transfer of dynamic training adaptations to an isometric task, rather than an absence of neuromuscular adaptations. We have now expanded the Discussion section to explicitly address task specificity and how this methodological choice may have influenced the magnitude of the observed adaptations.

Added to Discussion section: Additionally, task specificity may also have influenced the temporal pattern of the observed adaptations. The training program involved dynamic squats, while the testing task was an isometric leg-press. It is well-known that strength gains and the neural adaptations are specific to the training modality (as velocity and load) and contraction type (19, 68, 69). Dynamic resistance training has been shown to transfer only partially to isometric testing, with reduced improvements compared to task-matched training conditions (70). Therefore, the limited changes observed at intermediate time points may reflect incomplete transfer of training-induced adaptations to the isometric task rather than an absence of neuromuscular adaptations. This consideration is especially relevant when considering intermuscular coordination which is highly task dependent.

5.Interpretation of RF Activation

?The finding of decreased rectus femoris (RF) activation is intriguing but counterintuitive. The explanation provided (biarticular function, efficiency) is plausible but speculative. It would strengthen the manuscript to compare with existing studies involving biarticular muscles in multi-joint tasks.

Response: We agree that the observed decrease in rectus femoris (RF) activation is intriguing and that the physiological should be grounded in existing evidence considering the functional role of biarticular muscles. Recent works show that biarticular muscles such as RF are regionally recruited concerning the demands of the task. RF often contributes less the monoarticular portions of the quadriceps during multi-joint lower limb extension tasks and presents a spatially heterogeneous activation along its length depending on the emphasis of the task is hip flexion or knee extension. Moreover, single-joint knee extension tasks appear to preferentially recruit RF, while multi-joint tasks recruit the monoarticular portions of the quadriceps more strongly. Additionally, our recent work observed a trend for a reduced RER of RF in trained individuals compared to untrained ones, with different responses to fatigue in the two groups, which supports the notion that training status and the task influence the patterns of activation of the RF. These findings together suggest that decreases in RF activation during dynamic multi-joint training reflect adaptive coordination patterns. We have now expanded the Discussion section to integrate the recent evidence concerning RF biarticular behaviour.

Reformulated/Added to Discussion section: Indeed, the decreased activation of RF observed in the present study, although unexpected, aligns with recent evidence concerning the role of biarticular muscles, may be related with better efficiency of RF activation as a biarticular muscle, agonist of knee extension and antagonist of hip extension. First, studies using high-density surface EMG have shown that RF is regionally and task-dependently activated (79, 80), also depending on hip flexion angle (81). Second, the activity of RF during closed chain exercises has been demonstrated to be lower when compared to open chain exercises (76, 82). Third, during multi-joint exercises combining knee extension and hip extension, as the leg-press, RF activation is decreased when compared to single-joint knee extension, which may contribute to increased hip extension torque (76). Consequently, it has been shown that single-joint knee extension training is preferential to enhance RF activation rather than multi-joint exercises (83). Therefore, a reduction in RF RER after squat training may reflect a more efficient distribution of the neural drive during the task, reducing RF early activation when knee extension is combined with hip extension, and enhancing coordination with the monoarticular portions of the quadriceps. Nevertheless, recent work from our research group presented a trend for reduced RER of RF in highly trained individuals when compared to untrained ones, with different fatigue-related responses between groups (with the trained individuals increasing and the untrained individuals decreasing RF activation after fatigue) (84). These findings further support that RF activation patterns are modified with training experience, and therefore the observed reduction in RF RER may reflect a task-specific adaptation to optimize the intermuscular coordination patterns. Future studies using high-density surface EMG should explore the regional adaptations in RF activation to confirm if the observed activation results from a redistribution of the neural drive or an overall neural input to the muscle.

6.Figures & Tables

?Figures (especially IMC data) are difficult to interpret. Please ensure axis labels, frequency bands, and effect sizes are clearly presented.

Response: We thank the reviewer for this comment. We believe that the current figures already provide the necessary information to interpret the IMC data, including clearly labelled axes (frequency and coherence), consistent scaling across panels, and the indication of statistically significant coherence thresholds. Frequency-specific effects are explicitly quantified and statistically tested in the Results section rather than encoded visually in the figures, in order to avoid overloading the graphical presentation. For clarity, we have slightly refined the figure caption to better guide the reader through the interpretation of the IMC spectra.

?Table 1 should include confidence intervals in addition to means and SDs.

Response: We appreciate this su

---

## [Decision Letter · Decision Letter 1]

20 Mar 2026

Muscle activation and intermuscular coordination adaptations to early strength training during maximal force production

PONE-D-25-32197R1

Dear Dr. Santos,

We’re pleased to inform you that your manuscript has been judged scientifically suitable for publication and will be formally accepted for publication once it meets all outstanding technical requirements.

Kind regards,

Charlie M. Waugh

Academic Editor

PLOS One

Additional Editor Comments (optional):

Reviewers' comments:

Reviewer's Responses to Questions

**Comments to the Author**

Reviewer #2: All comments have been addressed

2. Is the manuscript technically sound, and do the data support the conclusions?

Reviewer #2: Yes

3. Has the statistical analysis been performed appropriately and rigorously?

Reviewer #2: Yes

4. Have the authors made all data underlying the findings in their manuscript fully available?

Reviewer #2: (No Response)

5. Is the manuscript presented in an intelligible fashion and written in standard English?

Reviewer #2: Yes

Reviewer #2: The authors have addressed my comments and concerns. I have no further comments or suggestions. Thank you.

.

Reviewer #2: No

---

## [Editor Report · Acceptance letter]

PONE-D-25-32197R1

PLOS One

Dear Dr. Santos,

I'm pleased to inform you that your manuscript has been deemed suitable for publication in PLOS One. Congratulations! Your manuscript is now being handed over to our production team.

Kind regards,

on behalf of

Dr. Charlie M. Waugh

Academic Editor

PLOS One